# Generation of a versatile BiFC ORFeome library for analyzing protein–protein interactions in live *Drosophila*

Johannes Bischof[1†], Marilyne Duffraisse[2†], Edy Furger[1†], Leiore Ajuria[2†], Guillaume Giraud[2], Solene Vanderperre[2], Rachel Paul[2], Mikael Björklund[3], Damien Ahr[2], Alexis W Ahmed[2], Lionel Spinelli[4], Christine Brun[4,5], Konrad Basler[1], Samir Merabet[2*]

[1]Institute of Molecular Life Sciences, University of Zurich, Zurich, Switzerland; [2]IGFL, CNRS, ENS Lyon, Lyon, France; [3]Zhejiang University-University of Edinburgh Institute, Zhejiang University, Haining, China; [4]INSERM, Aix-Marseille Université, Marseille, France; [5]TAGC, Centre National de la Recherche Scientifique, Marseille, France

**Abstract** Transcription factors achieve specificity by establishing intricate interaction networks that will change depending on the cell context. Capturing these interactions in live condition is however a challenging issue that requires sensitive and non-invasive methods.
We present a set of fly lines, called 'multicolor BiFC library', which covers most of the *Drosophila* transcription factors for performing Bimolecular Fluorescence Complementation (BiFC). The multicolor BiFC library can be used to probe two different binary interactions simultaneously and is compatible for large-scale interaction screens. The library can also be coupled with established *Drosophila* genetic resources to analyze interactions in the developmentally relevant expression domain of each protein partner. We provide proof of principle experiments of these various applications, using Hox proteins in the live *Drosophila* embryo as a case study. Overall this novel collection of ready-to-use fly lines constitutes an unprecedented genetic toolbox for the identification and analysis of protein-protein interactions in vivo.
DOI: https://doi.org/10.7554/eLife.38853.001

*For correspondence:
samir.merabet@ens-lyon.fr

†These authors contributed equally to this work

## Introduction

Proteins are distributed in various compartments within the cell, acting in a crowded environment and establishing hundreds of molecular contacts that will eventually dictate cellular function. These molecular contacts are often highly dynamic, may be of weak affinity, and depend on the cell context. Capturing these versatile interactions is therefore a key challenge to better understand the molecular cues underlying protein function in vivo.

Two types of experimental strategies are classically used for high-throughput screening of protein-protein interactions (PPIs): yeast two-hybrid (Y2H) and tandem affinity purification coupled to mass spectrometry (TAP-MS). Y2H detects PPIs in an automatable way in live yeast cells, while TAP-MS is based on co-immunoprecipitation (co-IP) and subsequent MS analysis of the different constituents of the complex (see [*Mann et al., 2013*; *Cusick et al., 2005*] for review). Recent developments such as proximity based biotinylation (BioID) have significantly increased the sensitivity of the TAP-MS approach, allowing capturing low affinity PPIs from a small number of cells (*Varnaité and MacNeill, 2016*). Despite their wide range of applications, Y2H and TAP-MS still have several drawbacks. For example, Y2H does not reproduce the plant or animal cell environment, and each revealed interaction needs thus to be validated in the relevant physiological context afterwards. TAP-MS can be

performed in the appropriate cell type, but the extraction protocol requires experimental conditions (especially for cell fixation and/or cell lysis) that are often not neutral to the integrity of endogenous PPIs. Finally, BioID allows interaction analyses without fixation, but may capture proteins promiscuously based on proximity rather than direct physical interactions, which could yield false positives.

In addition to these high-throughput approaches, PPIs can also be analyzed at a low-scale level, for example to validate the interaction status of one protein with few candidate partners. Co-IP followed by western blot is classically used for this purpose. However, this approach requires tools that are not systematically available, such as good antibodies or tagged-constructs. Co-IP experiments may also be poorly sensitive, detecting mainly interactions that will resist cell lysis and purification conditions.

An alternative and more sensitive approach is in situ proximity ligation assay (PLA), which provides a direct readout of the candidate PPI in the cell (*Bagchi et al., 2015*). A major limitation of PLA is the need of good antibodies against both interacting proteins. In addition, PLA works on fixed material, which is not neutral for PPIs.

Among the few methods that are compatible for PPI analysis in live conditions is Förster (or Fluorescence) resonance energy transfer (FRET), which relies on the transfer of a virtual photon between two fluorescent chromophores upon excitation. This transfer occurs within a small distance (less than 10 nm) and can thus be used to validate the close proximity between two candidate proteins, or to capture a conformational change (*Piston and Kremers, 2007*). FRET requires high level of protein expression and dedicated interfaces to interpret the few emitted signals with confidence and therefore cannot be used for large-scale applications.

By contrast, Bimolecular Fluorescence Complementation (BiFC) appears much more convenient since it is based on a visible fluorescent signal. BiFC relies on the property of monomeric fluorescent proteins to be reconstituted from two sub-fragments upon spatial proximity (in a similar range of distance as FRET). This method has been used in different plant and animal model systems and with various types of proteins (*Kerppola, 2008*; *Kodama and Hu, 2012*; *Miller et al., 2015*). In particular, recent work has established experimental parameters for performing BiFC in the live *Drosophila* embryo (*Hudry et al., 2011*), and the method was coupled to a candidate gene approach to identify new interacting partners of *Drosophila* Hox proteins (*Baëza et al., 2015*).

Moreover, the bright intrinsic fluorescence of BiFC allows analysis of PPIs using commonly available fluorescent microscopes and with normal protein expression levels. Originally established with the Green Fluorescent Protein (GFP [*Ghosh et al., 2000*]), BiFC has by now been developed with various GFP-derivatives such as the YFP, Venus or Cerulean proteins (*Hu et al., 2002*; *Hu and Kerppola, 2003*; *Shyu et al., 2006*). BiFC has also been established with other types of monomeric fluorescent proteins, including red fluorescent variants like mRFP1 (*Jach et al., 2006*) or mCherry (*Fan et al., 2008*), and more recently the near infrared fluorescent protein iRFP (*Chen et al., 2015*). In all cases, the complementation between the two sub-fragments of the fluorescent protein induces the formation of covalent junctions, leading to a stabilization of the protein complex. While this property forbids monitoring temporal dynamics of PPIs, this practically irreversible nature of the complementation allows detection of weak and otherwise transient PPIs, making BiFC a very sensitive approach for studying PPIs in vivo. BiFC has also been used in several high throughput approaches in yeast (*Sung et al., 2013*), plant (*Lee et al., 2012*) and mammalian cells (*Lee et al., 2011*), or for drug discovery against a specific PPI (*Poe et al., 2014*; *Dai et al., 2013*), demonstrating its suitability for large-scale applications.

Here, we present a genetic repertoire covering 450 *Drosophila* transcription factors (TFs) (corresponding to around 65% of annotated TFs) for performing BiFC in a tissue- and developmental stage-specific manner in vivo. This genetic repertoire is called multicolor BiFC library and consists of a collection of fly lines that complement the previously established FlyORF collection (https://flyorf. ch/ and [*Bischof et al., 2013*]). The multicolor BiFC library is continually updated and aims at covering all *Drosophila* TFs in the near future. We provide proof of concept experiments showing the suitability of the BiFC FlyORF library for performing either large-scale interaction screens or for analyzing individual PPIs. This collection of fly lines constitutes the first genetic toolbox for analyzing thousands of different PPIs in a live animal organism, opening new avenues for understanding molecular properties of protein interaction networks.

## Results

### Generation of a fly library containing Gal4-inducible ORFs compatible with Venus-based BiFC

560 open-reading frames (ORFs) among the 3000 actually present in the FlyORF library code for TFs. These ORFs are under the control of upstream activation sequences (UAS sites) and fused in frame to a hemagglutinin tag (3xHA) sequence at their 3' end (*Bischof et al., 2013*). The ORFs are flanked by distinct FRT sites that can be used to replace the promoter sequence and/or the 3xHA tag region by any other sequence of choice upon FLP/FRT-mediated recombination in vivo (*Bischof et al., 2013*). In particular, two swapping fly lines have been generated to replace the C-terminal 3xHA-tag by sequences coding for the N- or C-terminal fragment of the Venus fluorescent protein (fragments respectively called VN and VC hereafter in the manuscript [*Bischof et al., 2013*]). These two fragments, when attached to proteins that are co-expressed, are able to complement upon spatial proximity, allowing assessment of the interaction by BiFC (*Figure 1A*). A proof of principle experiment between two known interacting partners present in the FlyORF library proved the efficiency of the swapping and the specificity of the Venus-based BiFC in the *Drosophila* wing imaginal disc (*Bischof et al., 2013*).

Experimental parameters for performing Venus-based BiFC have also been established in different tissues of the live *Drosophila* embryo (*Hudry et al., 2011*). In particular, several controls showed

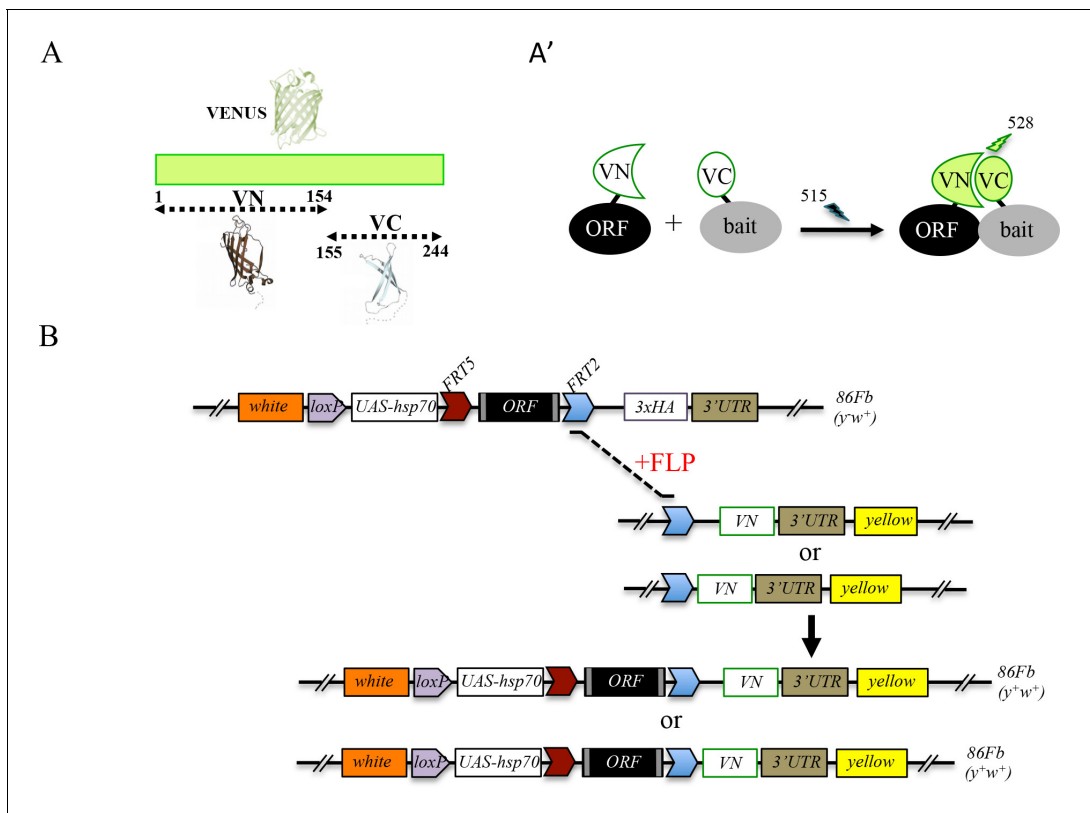

**Figure 1.** Generation of a Gal4 inducible library compatible with Venus-based BiFC in *Drosophila*. (A-A') Principle of the Venus-based BiFC between a candidate ORF (Open Reading Frame) and a bait protein fused to the N- (VN) or C-terminal (VC) fragment of Venus, respectively. Excitation and emission wavelengths are indicated. (B) Principles of Flippase (FLP)/FRT-mediated recombination to swap the C-terminal 3xHA tag of the ORF with the original VN or new VN-short tag line. Genetic crosses and selection procedure are described in (*Bischof et al., 2013*). Note that the UAS-ORF-HA and resulting UAS-ORF-VN are located on the third chromosome (*86Fb*). See also *Figure 1—figure supplement 1* and *Supplementary file 1*.

DOI: https://doi.org/10.7554/eLife.38853.002

The following figure supplement is available for figure 1:

**Figure supplement 1.** Comparison of Venus-based BiFC when using ORFs swapped with the original VN or VN-short tag line.

DOI: https://doi.org/10.7554/eLife.38853.003

that Venus-based BiFC could not occur under conditions where the interaction between the two candidate partners was disrupted. This was demonstrated by using mutant proteins (*Hudry et al., 2011*) or by coexpressing one partner that cannot complement (not fused to a Venus fragment) and thus competes against BiFC (*Baëza et al., 2015*).

Although very convenient for generating new fusions without additional fly transgenesis, swapping experiments require several generations, and therefore several weeks, before getting the desired BiFC-compatible fly line (*Bischof et al., 2014*). In order to introduce ready-to-use fly lines for performing Venus-based BiFC, we systematically exchanged the 3xHA tag in a number of TF lines of the FlyORF library for the VN tag (*Figure 1B*). These first rounds of swapping experiments led to a collection of 136 ORF-VN fly lines (*Supplementary file 1*, third column). We further generated an alternative VN-swapping fly line that allows fusing the ORF to the Venus fragment, however with a shorter linker region between ORF and VN (*Figure 1B* and *Figure 1—figure supplement 1*, see also Materials and methods). Reducing the length of this region could potentially diminish the risk of revealing indirect PPIs in vivo. A series of pilot tests confirmed that the new VN-short tag fly line is suitable for swapping and BiFC experiments (*Figure 1—figure supplement 1*). This new swapping fly line is now systematically used for generating additional ready-to-use ORF-VN tagged fly lines. To date, 74 ORFs have been fused to the VN fragment with the short linker region (*Supplementary file 1*, fourth column). In addition to the previously generated fly lines (including those generated in (*Hudry et al., 2011*; *Baëza et al., 2015*), *Supplementary file 1*, fifth column), the multicolor BiFC library currently contains 232 different TFs fused to VN for doing Venus-based BiFC in *Drosophila*.

## Generation of a fly library containing Gal4-inducible ORFs compatible with bicolor BiFC

One interesting feature of BiFC is the opportunity of using fragments from various GFP-derived proteins to visualize two different PPIs in the same cell (*Hu and Kerppola, 2003*). In particular, it was shown that the C-terminal fragment of the blue fluorescent Cerulean protein (CC) could complement with either the N-terminal fragment of Venus (VN) or Cerulean (CN), giving rise to Venus or Cerulean-like fluorescent signals, respectively (Figure 2A-A' and [*Hu and Kerppola, 2003*]). Given that Cerulean-based BiFC was shown to be exploitable in the *Drosophila* embryo (*Hudry et al., 2011*), we decided to increase the versatility of the BiFC library by generating an additional set of fly lines that could be used for both Venus- and Cerulean-based BiFC (*Figure 2B*). This new UAS-inducible ORF-CC library was inserted on the second chromosome (see Materials and methods and *Figure 2—figure supplement 1*) and can be used to perform Venus- and Cerulean-based BiFC with any VN-(including VN fusion constructs of the library) and CN-fused bait protein, respectively. Fly lines for 326 different ORF-CC have been generated so far (*Supplementary file 2*).

To verify that the ORF-CC library is compatible with Cerulean- or Venus-based BiFC, we chose a set of six TFs that have already been analyzed as VN-fusion constructs with the Hox protein AbdominalA (AbdA) fused to the VC complementary fragment (Bagpipe, Bap; Empty spiracles, Ems; Knirps, Kni; Pangolin, Pan; Pox mesoderm, Poxm; Tinman, Tin), and whose interaction status was also validated by co-IP experiments (*Baëza et al., 2015*). Among those six TFs, all but Pan could interact with AbdA (*Baëza et al., 2015*). The same pool of TFs was here tested as ORF-CC fusions with VN-AbdA or CN-AbdA in two different tissues (epidermis and somatic mesoderm) and at two different developmental stages (stages 10 and 12) by using two different Gal4 drivers. Results showed that Venus and Cerulean fluorescent signals could easily be distinguished from the fluorescent background in the epidermis or mesoderm in the case of all tested TFs except Pan (*Figure 2C–D*). Moreover, competition tests with the corresponding HA-tagged ORFs validated the specificity of BiFC obtained between VN-AbdA and each candidate ORF-CC construct (*Figure 2—figure supplement 2*). This last experiment confirms that the ORF-3xHA fly lines from the original FlyORF library can be used to verify the specificity of BiFC signals. Altogether, these observations establish that the ORF-CC constructs of the multicolor BiFC library are compatible with either Cerulean- or Venus-based BiFC in different tissues of the live *Drosophila* embryo.

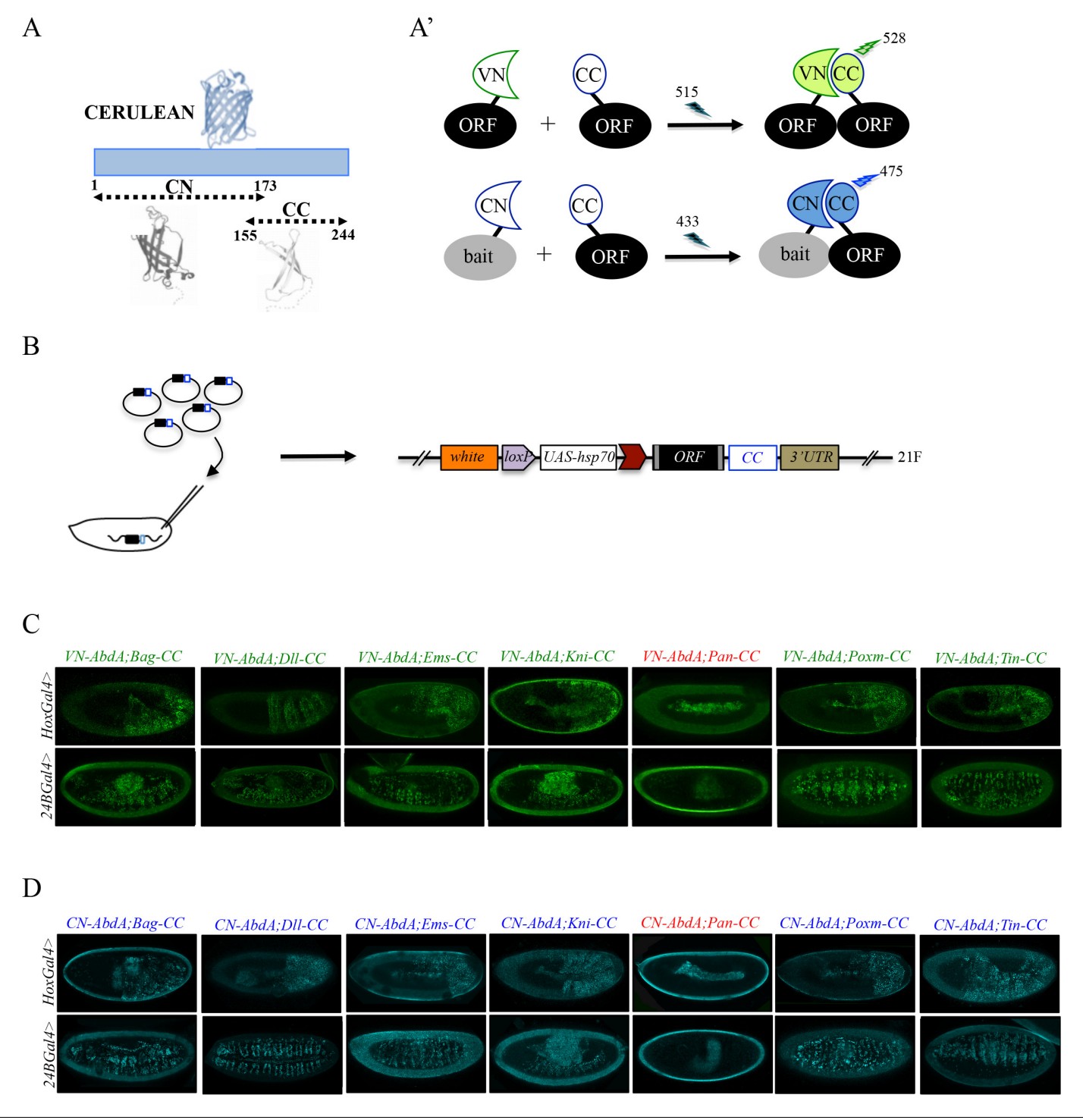

**Figure 2.** Generation of a Gal4 inducible library compatible with Venus- and Cerulean-based BiFC in *Drosophila*. (**A-A'**) Principle of bicolor BiFC by using the complementation property between the C-terminal fragment of the blue fluorescent protein Cerulean (CC) and the N-terminal fragment of Venus (VN) or Cerulean (CN). Excitation and emission wavelengths are indicated. (**B**) Principle of the generation of the UAS-ORF-CC library at the 21F genomic locus. See also Materials and methods. (**C-D**) Illustrative confocal captures of Venus-based BiFC obtained from different ORF-CCs and VN-AbdA (**C**) or CN-AbdA (**D**) interaction partners, as indicated. Fusion proteins are expressed with the *abdA-Gal4* (upper panels) or *24B-Gal4* (lower panels) driver and BiFC is observed in the epidermis (stage 10/11) or somatic mesoderm (stage 14), respectively. Note that the amnioserosa, the gut inside the embryo and the vitelline membrane around the embryo display strong autofluorescence. Absence of interaction with Pangolin (Pan) is highlighted in red. See also *Figure 2—figure supplements 1* and *2* and *Supplementary file 2* and *3*.

*Figure 2 continued on next page*

*Figure 2 continued*

DOI: https://doi.org/10.7554/eLife.38853.004

The following figure supplements are available for figure 2:

**Figure supplement 1.** The 21F and 86Fb attP sites lead to similar expression levels.

DOI: https://doi.org/10.7554/eLife.38853.005

**Figure supplement 2.** Co-expression of non-complementing ORF-3xHA can compete against BiFC signals obtained from ORF-CC and VN-AbdA constructs.

DOI: https://doi.org/10.7554/eLife.38853.006

## Using the multicolor BiFC library for large-scale interaction screens in live *Drosophila* embryos

The multicolor BiFC library currently covers 450 different *Drosophila* TFs (around 65% of annotated TFs), making a total of 579 fly lines due to several TFs fused with different VN and/or CC versions. Among the 450 TFs, 127 are available as VN fusion constructs, 221 as CC fusion constructs, and 105 as VN- and CC-fusion constructs (*Figure 3*). Together with the simplicity of genetic crosses and read-outs, it makes this library appropriate for large-scale interaction screens in *Drosophila*.

We provide a proof of concept by analyzing interaction properties of 260 TFs of the multicolor BiFC library with Ultrabithorax (Ubx) and AbdA (*Figure 4A*). These two Hox proteins have highly similar domains and motifs (*Figure 4—figure supplement 1*). They share a number of common functions during embryogenesis and can substitute for each other in several tissues, as noticed for example in the epidermis (*Gebelein et al., 2004*), trachea (*Merabet et al., 2005*) or ventral nerve cord (*Ahn et al., 2010*). Ubx and AbdA have also few distinct expression patterns that correlate with specific functions in the embryo. For example, the specification of oenocytes (*Brodu et al., 2002*), heart (*Ponzielli et al., 2002*) or gonadal mesoderm (*Riechmann et al., 1998*) is under the control of AbdA, a function that cannot be substituted by Ubx. Such exclusive functions are however rare during *Drosophila* embryogenesis, suggesting that most embryonic functions of Ubx and AbdA could rely on the interaction with a large number of common cofactors. In this context, we aimed at using the multicolor BiFC library to reveal interactions that could be specific to AbdA. The identification of such partners would validate the specificity and sensitivity of the multicolor BiFC library in the context of a large-scale interaction screen with two closely related bait proteins.

Among the 260 different TFs, 127 were analyzed in fusion with VN and 133 in fusion with CC (*Supplementary file 3*). Thirty-five TFs were tested in fusion with VN and CC to assess the influence of the fusion topology on BiFC results (*Supplementary file 8*, see also Discussion). Interactions were analyzed in the epidermis of stage 10 embryos, in the *Ubx-* or *abdA*-expression domain, by using *Ubx-Gal4* or *abdA-Gal4* driver, as previously described (*Baëza et al., 2015*).

The analysis showed that 62% (163/260) and 57% (149/260) of the tested TFs could interact with Ubx or AbdA, respectively (*Figure 4A–B*, *Figure 4—figure supplements 2–6* and *Supplementary file 4*). Among all positive interactions, 71,4% (130/182) are common to the Ubx and AbdA interactomes (*Figure 4—figure supplements 7–8*), consistent with the numerous overlapping functions between the two Hox proteins during embryogenesis. Interestingly, the homeodomain (HD) class of TFs appears significantly enriched in the Ubx (p value=0,041) and AbdA (p value=0,017) interactomes, suggesting that HD-containing TFs could constitute a privileged class of common cofactors (*Figure 4C and D* and *Figure 4—figure supplement 1*). We also looked at TFs that did not interact with the Hox proteins, constituting the so-called negatomes (*Smialowski et al., 2010*), but did not notice enrichment for any particular class of TFs (*Figure 4—figure supplements 1* and *7*).

Although Ubx and AbdA shared a number of common interactions, our BiFC screen also revealed interactions that were specific for only one of the two Hox proteins. These interactions represent 20% and 13% of their overall interactome, respectively (33 TFs for Ubx and 19 TFs for AbdA, *Figure 4—figure supplements 8* and *9*). This result confirms that the multicolor BiFC library can be used to capture specific interaction partners between two closely related bait proteins. Interestingly, bZIP-containing TFs are absent from the two specific interactomes (*Figure 4E and F*), suggesting that this class of TFs could be specifically dedicated to Ubx- and AbdA common functions.

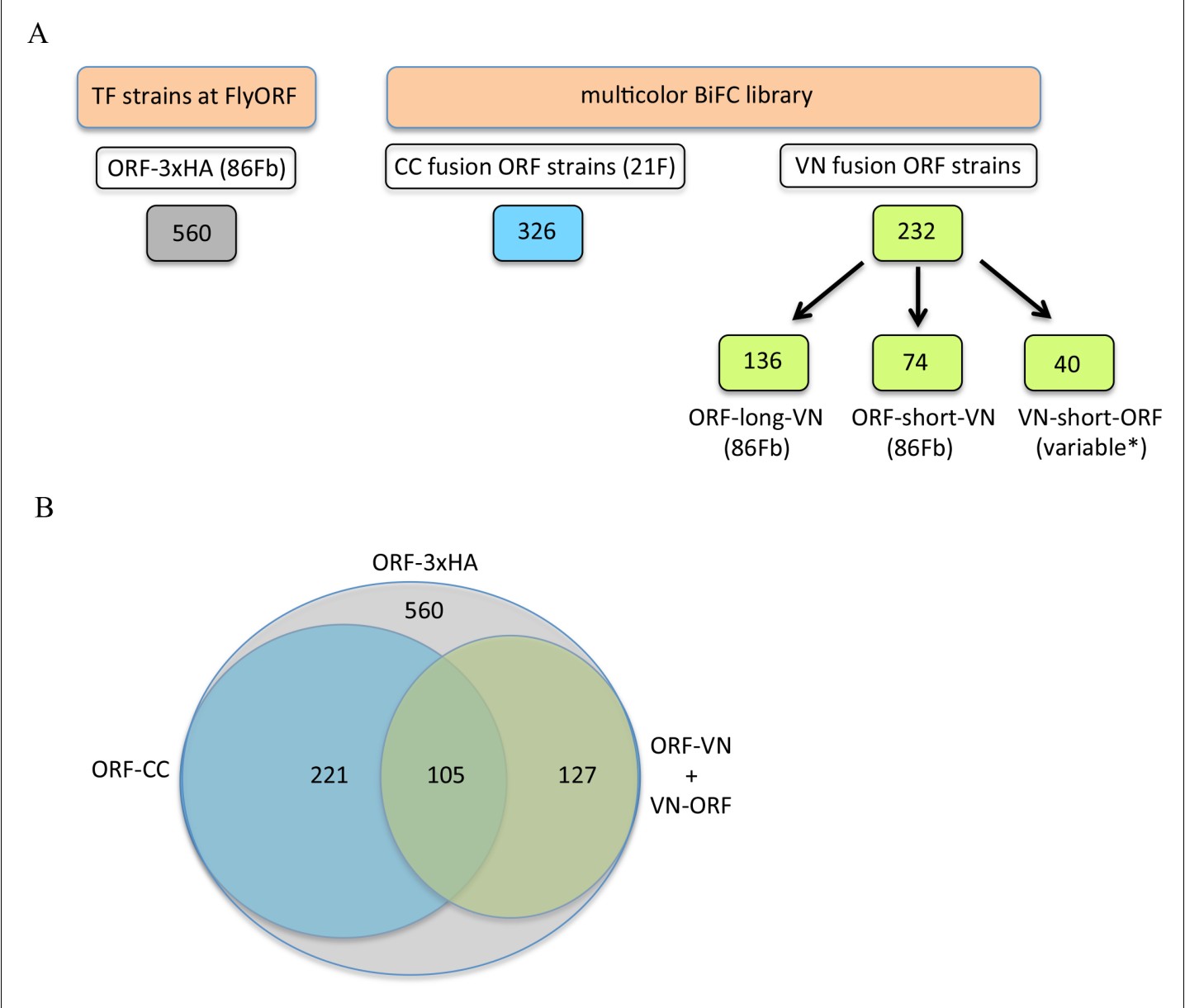

**Figure 3.** The TF-3xHA and multicolour BiFC libraries. (**A**) Number of transcription factors (TFs) tagged with three repeats of the hemagglutinin (3xHA) epitope at the 3' end and multicolour BiFC fly strains at FlyORF. Various insertion sites were used for the VN-short-ORF constructs (***Baëza et al., 2015***). Some of the 232 TF strains exist in more than one VN version (see also ***Supplementary file 1***). (**B**) Distribution of the multicolour BiFC fly lines compared to the TF-3xHA library.

DOI: https://doi.org/10.7554/eLife.38853.007

Overall, the BiFC screen shows that the two Hox proteins could interact with a surprisingly high number of various types of TFs. A similar conclusion (with 41% of positive interactions) was obtained from a previous candidate gene screen based on competitive BiFC with a set of 80 TFs (***Baëza et al., 2015***). This high interaction potential of Hox proteins could be explained by their numerous functions during embryogenesis and the extreme sensitivity of BiFC (linked to the fluorescent signal and UAS/Gal4 expression system).

The BiFC screen was performed in the *Ubx*- or *abdA*-expression domain by expressing the TFs using *Ubx-Gal4* or *abdA-Gal4* driver. True positive interactions are expected to occur between TFs that are co-expressed in the same spatial expression domain. To test this, we considered the extent of the overlap between the expression domain of Ubx or AbdA and each TF (as the ratio between

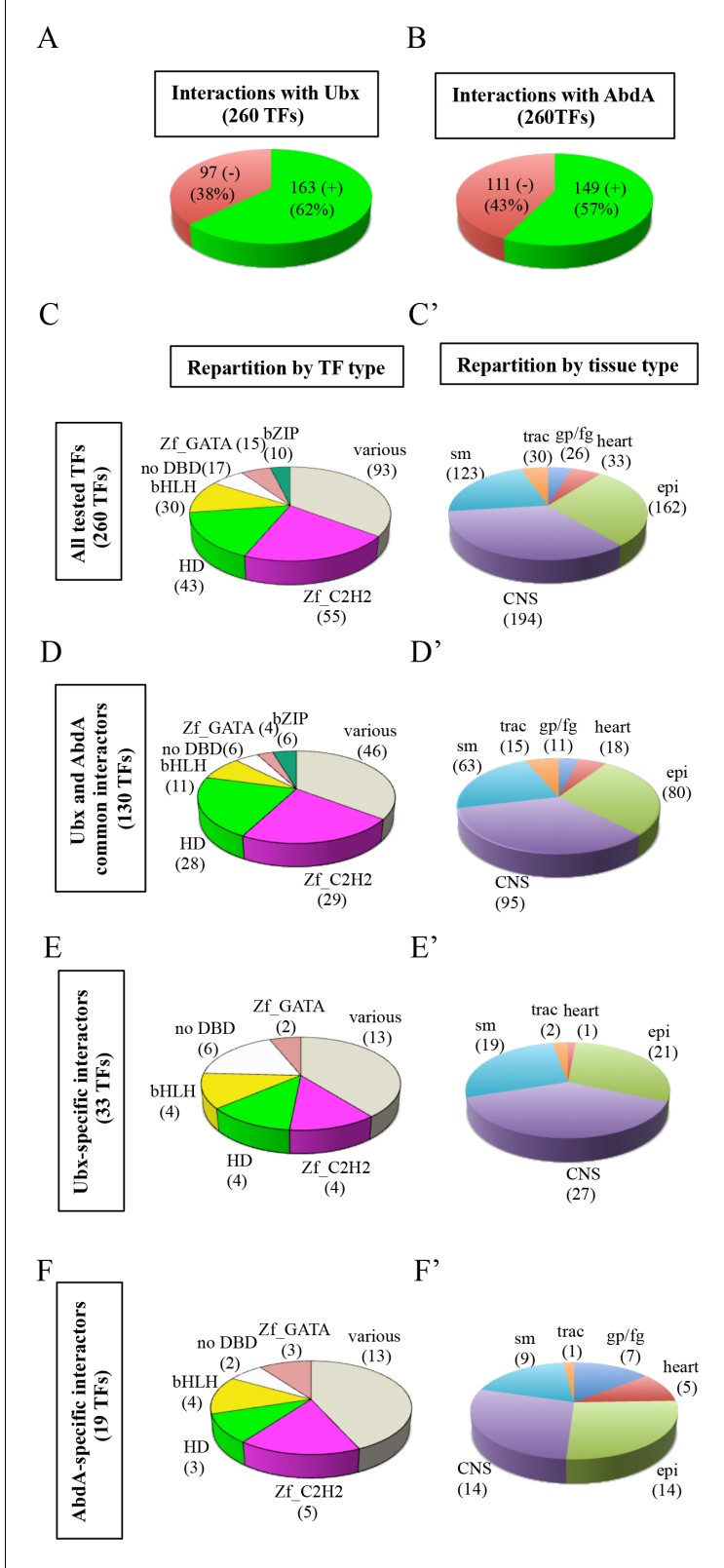

**Figure 4.** Using the multicolor BiFC library for a large-scale interaction screen with Ubx and AbdA in the live *Drosophila* embryo. (**A**) Number of TFs that were positive (green) or negative (red) with Ubx. (**B**) Number of TFs that were positive (green) or negative (red) with AbdA. (**C**) Repartition of the different families among the 260 TFs tested with the Hox proteins. A specific color code is attributed to each TF family. Families with the highest

*Figure 4 continued on next page*

*Figure 4 continued*

number of tested TFs are represented (Zinc fingers C2H2, Zf_C2H2; homeodomain, HD; basic helix-loop-helix, bHLH; no DNA-binding domain, no DBD; zinc fingers GATA, Zf_GATA; basic leucine zipper, bZIP). 36 other different families containing one to eight TF representatives are present in the 'various' category. (D) Repartition of the TF families among the 130 positive interactions common to Ubx and AbdA. Note that the HD family is slightly enriched in this interactome. (E) Repartition of the TF families among the 33 Ubx-specific interactions. (F) Repartition of the TF families among the 19 AbdA-specific interactions. Note the absence of bZIP representatives in the Ubx- and AbdA-specific interactomes. (C') Repartition of the expression profile of the 260 tested TFs in six different embryonic tissues: the somatic mesoderm (sm), trachea (trac), gonad primordium/fat body (gp/fb), heart, epidermis (epi) and central nervous system (CNS). Most of these TFs are expressed in several embryonic tissues. (D') Tissue-type repartition of the expression profile of the 130 Ubx- and AbdA-positive interactors. (E') Tissue-type repartition of the expression profile of the 33 Ubx-specific interactors. Note the absence of TFs expressed in the gp/fb. (F') Tissue-type repartition of the expression profile of the 19 AbdA-specific interactors. Note the specific enrichment of TFs expressed in the gp/fb. See also *Figure 4—figure supplements 1–10* and *Supplementary file 4* and *5*.

DOI: https://doi.org/10.7554/eLife.38853.008

The following figure supplements are available for figure 4:

**Figure supplement 1.** Interactomes and negatomes of Ubx and AbdA with the 260 tested TFs.
DOI: https://doi.org/10.7554/eLife.38853.009

**Figure supplement 2.** Illustrative confocal captures of Venus-based BiFC between ORF-VN and VC-Ubx, as indicated.
DOI: https://doi.org/10.7554/eLife.38853.010

**Figure supplement 3.** Illustrative confocal captures of Venus-based BiFC between ORF-CC and VN-Ubx, as indicated.
DOI: https://doi.org/10.7554/eLife.38853.011

**Figure supplement 4.** Illustrative confocal captures of Venus-based BiFC between ORF-VN and VC-AbdA, as indicated.
DOI: https://doi.org/10.7554/eLife.38853.012

**Figure supplement 5.** Illustrative confocal captures of Venus-based BiFC between ORF-CC and VN-AbdA, as indicated.
DOI: https://doi.org/10.7554/eLife.38853.013

**Figure supplement 6.** Illustrative confocal captures of Cerulean-based BiFC between ORF-CC and CN-AbdA, as indicated.
DOI: https://doi.org/10.7554/eLife.38853.014

**Figure supplement 7.** Representation of the common interactome and negatome between Ubx and AbdA.
DOI: https://doi.org/10.7554/eLife.38853.015

**Figure supplement 8.** Comparison between Interactomes.
DOI: https://doi.org/10.7554/eLife.38853.016

**Figure supplement 9.** Representation of the Ubx- (A) and AbdA-specific (B) interactomes.
DOI: https://doi.org/10.7554/eLife.38853.017

**Figure supplement 10.** Analysis of the co-expression and interaction status of Ubx (A) or AbdA (B) and the positive ORFs.
DOI: https://doi.org/10.7554/eLife.38853.018

the number of tissues in which the TF and Ubx or AbdA are co-expressed and the total number of tissues composing the TF expression domain during embryogenesis). To this end, we used annotations from (*Hammonds et al., 2013*) and the Flybase database to assign the expression status of Ubx, AbdA and each TF in 25 different developmental contexts (*Supplementary file 4*). This analysis showed that the extent of co-expression is significantly higher when TFs and Hox protein interactions are detected with BiFC than when they are not (Wilcoxon test, pvalue = $1.10^{-2}$ and $5.10^{-5}$ for Ubx and AbdA, respectively: *Figure 4—figure supplement 10*). Thus, the BiFC screen reveals more frequently interactions between TFs and Hox proteins when they are co-expressed during embryogenesis.

Further analysis of interactomes in major embryonic tissues (*Figure 4C', D', E' and F'*) revealed that the AbdA-specific interactome was enriched in TFs expressed in the fat body/gonad primordium (p value=$8.10^{-4}$) when compared to all positive interactions (*Figure 4F'*, encircled TFs in *Figure 4—figure supplement 9*, and *Supplementary file 5*). In contrast, TFs expressed in this tissue

are not present in the Ubx-specific interactome (*Figure 4E'*). This observation is consistent with the specific role of AbdA in the gonad primordium (*Moore et al., 1998*). Another TF, Spalt major (Salm), is also found in the AbdA-specific interactome (encircled by a dotted-line in *Figure 4—figure supplement 9*). Among other functions, Salm is important for oenocyte specification (*Elstob et al., 2001*), a role also specifically ensured by AbdA during embryogenesis (*Brodu et al., 2002*). A significant enrichment (p value=$5.10^{-2}$) is also observed in the dorsal vessel (*Figure 4F'*, TFs annotated with a star in *Figure 4—figure supplement 9*, and *Supplementary file 5*), which again coincides with a specific function of AbdA in particular for the differentiation of ostia and heart beating activity (*Ponzielli et al., 2002*). By comparison, TFs expressed in tissues where Ubx and AbdA have redundant/common functions show no significant enrichment in the AbdA-specific interactome (i.e epidermis, CNS, somatic mesoderm and trachea: *Figure 4D'* and F' and *Supplementary file 5*). Together these observations underline that the multicolor BiFC library is efficient in revealing relevant candidate cofactors involved in AbdA-and tissue-specific functions.

## Validating BiFC observations by a functional genetics approach in haltere primordium

Since BiFC revealed a number of potential interactions, we asked whether hits from the large-scale interaction screen could somehow be confirmed to contribute to Hox protein function. To this end, we searched for a sensitive genetic background where any subtle modification in Hox protein function could lead to a quantifiable phenotype. In this context, the loss of a Hox cofactor could affect the sensitized Hox protein function and lead to a stronger phenotype. No sensitive Hox-dependent phenotypes are known in the embryo, but several exist in the adult, like eye reduction (*Plaza et al., 2001*) or antenna-to-leg transformation (*Plaza et al., 2008*). These phenotypes rely however on ectopic expression of the Hox product and are therefore not ideal for assessing the role of a candidate cofactor in the normal developmental context. To circumvent this problem, we considered the haltere-to-wing phenotype, which results from the specific loss of Ubx in the haltere primordia (*Lewis, 1978*). A particular combination of *Ubx* mutant alleles has more recently been used to measure the ability of different Ubx isoforms to rescue haltere formation (*de Navas et al., 2011*). Importantly, the haltere-to-wing phenotype is sensitive to the dose of Ubx, and removing one copy of the Hox gene is sufficient to induce the formation of few small hairs that are normally found in the wing margin, revealing a weak haltere-to-wing transformation phenotype (*Figure 5A–B*). In this context, RNAi against *Ubx* is sufficient to induce a strong haltere-to-wing transformation, with the formation of numerous wing-like hairs and a flattened wing-like shape (*Figure 5A–B*). This result highlights that the heterozygote *Ubx* mutant phenotype can be increased when *Ubx* function is affected. We decided to use this sensitive background for measuring the role of TFs tested in our BiFC screen (see Materials and methods). The rational was that affecting the expression of a positive TF acting as a Ubx cofactor in the haltere primordium should increase the haltere-to-wing phenotype. Reversely, a negative TF should have no effect.

TFs were selected based on their known expression and/or function in the haltere (Flybase database, (*Schertel et al., 2015*) and *Supplementary file 6*). The resulting 40 TFs (among all the 260 that were tested in the embryo BiFC screen) were assayed using RNAi experiments in *Ubx* heterozygous mutant haltere discs. Slightly more than half of the tested TFs (23/41: *Supplementary file 6*) did not increase the phenotype of the heterozygous *Ubx* mutant upon RNAi (as illustrated with RNAi against *absent_small_or_homeotic_discs_2* (*ash2*) in *Figure 5A–B*). The large majority of those TFs (18/23) were classified as BiFC negative with Ubx in the embryo, confirming that BiFC was specific enough not to identify these as interacting partners of Ubx (*Figure 5C* and *Supplementary file 6*). In addition, we cannot exclude the possibility that the five other TFs that were BiFC positive could act as Ubx cofactors in another developmental context. Among the 17 remaining TFs, eight enhanced the phenotype upon RNAi (*Supplementary file 6*). The phenotype enhancement was not as strong as with the RNAi against Ubx, and consisted in the appearance of several wing-like hairs in the haltere (as illustrated with RNAi against *homothorax* (*hth*) in *Figure 5A–B*). 7/8 of those TFs scoring positively in the RNAi assay were also BiFC-positive in the embryo, highlighting that Ubx cofactors were efficiently captured in the BiFC interaction screen (*Figure 5D* and *Supplementary file 6*). It is also worth mentioning that the TF that gave the strongest phenotype upon RNAi, Hth, displays enriched binding adjacent to Ubx binding sites genome wide in the haltere tissue (*Choo et al., 2011*). This suggests that Hth could constitute a crucial cofactor for Ubx in the haltere specification

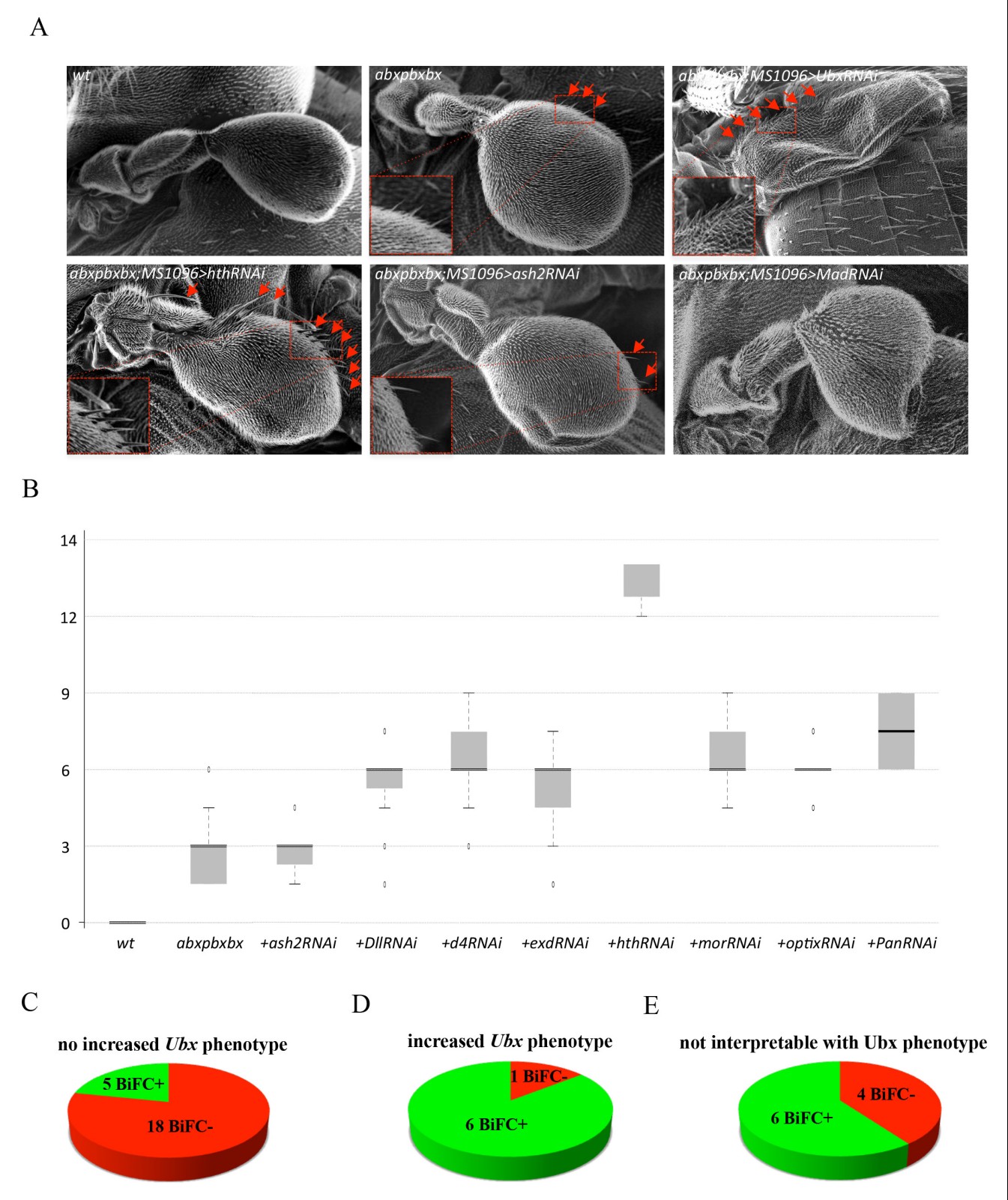

**Figure 5.** Functional genetics validates BiFC observations with Ubx in haltere primordium. (**A**) Scanning electron microscopy of haltere phenotypes in the different genetic backgrounds, as indicated. Compared to wild-type, halteres of individuals heterozygous for the Hox regulatory mutation *abxpbxbx* have ectopic short wing-like hairs (arrows and zoom in/enlargement in B). This phenotype is increased when affecting *Ubx* expression upon expression of RNAi or when expressing a RNAi against a TF (although to a lesser extent: shown here for *Homothorax, Hth*) that could be required for Ubx function
*Figure 5 continued on next page*

*Figure 5 continued*

(arrows and enlargements). In contrast, expression of RNAi against a TF that is not required for Ubx function (as shown for *absent_small_or_homeotic_discs _2, ash2*) does not increase the phenotype. The expression of RNAi against TFs can also affect more globally the haltere (and wing) formation (as shown for Mad), which is difficult to interpret in term of homoeotic transformation and therefore with regard to a potential Ubx cofactor function. (B) Box plot statistical quantification of the haltere-to-wing transformations in the different genetic backgrounds, as indicated. Quantification was performed by counting the number of ectopic wing-like hairs formed at the edge of the haltere and on the hinge. The phenotype induced by the *Ubx*RNAi was voluntary not included since it corresponds to an almost complete haltere-to-wing transformation. (C-E) Diagrams showing the distribution of TFs that were BiFC positive (green) and negative (red) with Ubx in the different cases (not increased haltere phenotype (C); increased haltere phenotype (D); not interpretable (E)). See also *Supplementary file 6*.

DOI: https://doi.org/10.7554/eLife.38853.019

program. Finally, RNAi against 10 TFs led to malformations that were difficult to interpret with regard to a potential role as Ubx cofactor since the morphological defects could also be independent of a Ubx cofactor function. An illustrative phenotype is given with a RNAi against the TF Mad (*Figure 5A*). Because of this ambiguous putative role, these TFs were not further considered (*Figure 5E* and black boxes in *Supplementary file 6*).

Overall, the haltere sensitized genetic background highlighted that 80% (25/31) of the selected TFs that give an interpretable phenotype were correlated to the BiFC interaction status. Thus, results obtained from the BiFC screen in the embryo could be reproduced at the functional level in another developmental context.

## Using the multicolor BiFC library for analyzing two different PPIs in the same embryo

In addition to large-scale interaction screens, the multicolor BiFC library aims at providing new perspectives for the analysis of individual PPIs in vivo. In particular, the ORF-CC library allows performing Venus- and Cerulean-based BiFC, therefore analyzing two different PPIs in the same embryo. This so-called 'multicolor' property was established in live cells (*Hu and Kerppola, 2003*) and recently extended for use in live *Drosophila* embryos (*Hudry et al., 2011*).

Here we asked whether the multicolor BiFC library could be used for analyzing two different PPIs simultaneously. This potential was more precisely examined in the context of a well-known partnership between AbdA and the cofactor Extradenticle (Exd). Exd belongs to the PBC class of TALE (Three Amino Acids Loop Extention) TFs (*Mukherjee and Bürglin, 2007*). PBC proteins constitute an ancestral and generic class of Hox cofactors, and the activity of Hox-PBC complexes has been described in numerous developmental contexts (*Mann et al., 2009*). In particular, the partnership between AbdA and Exd is involved in early (e.g., patterning) and late (e.g., gut morphogenesis) developmental processes during *Drosophila* embryogenesis, suggesting that a number of TFs could interact with the two proteins, potentially participating in the activity of AbdA/Exd complexes (*Figure 6A*).

In order to identify such partners, we considered a set of 37 TFs fused to VN and previously described to be positive with VC-AbdA ( (*Baëza et al., 2015*) and this work: *Supplementary file 7*). This set of TFs was tested with the VC-Exd fusion protein and BiFC was analysed in the epidermis of stage 10 embryos with the *abdA-Gal4* driver, as previously described (*Duffraisse et al., 2014*). Results showed that 22/37 TFs were positive with Exd (*Figure 4—figure supplement 8*, *Figure 6—figure supplements 1* and *2* and *Supplementary file 7*), highlighting that a small majority of AbdA-positive interactions (59%) was also positive with Exd, but also that an important proportion of AbdA interactions (41%) might occur independently of the partnership with Exd in vivo.

We next selected four TFs (Tin, CtBP, Distalless (Dll) and Bagpipe (Bag)) that were positive with AbdA and Exd and present in the ORF-CC library to perform bicolour BiFC. An additional TF, Empty spiracles (Ems), which was positive with AbdA ( (*Baëza et al., 2015*) and *Figure 4—figure supplement 3*) but negative with Exd (*Figure 6—figure supplement 2*), was also considered for the specificity control. The corresponding ORF-CC fusions were expressed together with CN-AbdA, VN-Exd and a mCherry reporter to trace Gal4-expressing cells in the embryo. Analysis with Ems-CC confirmed the respective positive or negative interaction status with AbdA or Exd, validating the specificity of BiFC upon the co-expression of the three fusion proteins in the same embryo (*Figure 6B*). Analysis with the four other TFs showed that the majority of expressing nuclei (as assessed with the

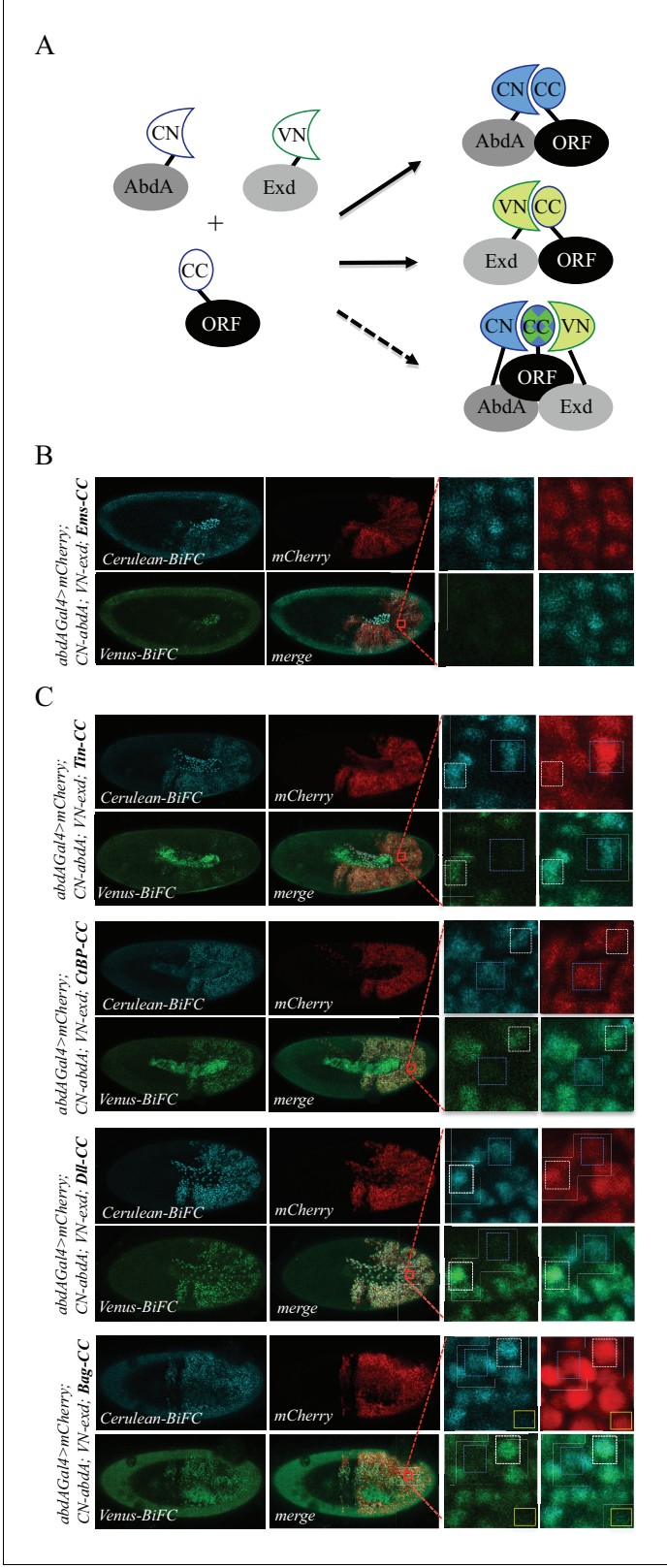

**Figure 6.** Using the multicolour BiFC library for analysing two different interactions in the same embryo. (**A**) Principle of the bicolour BiFC. The AbdA and Extradenticle (Exd) cofactor are respectively fused to the CN or VN fragment, which can complement with the CC fragment of a co-expressed ORF when interaction occurs. The simultaneous expression of the three fusion proteins allows assessing Venus- and Cerulean-based BiFC in the

*Figure 6 continued*

same cell. Bicolour BiFC results from the interaction of the ORF-CC with both CN-AbdA and VN-Exd, thus revealing two binary interactions simultaneously. BiFC could result from interactions occurring in two independent complexes but potentially also in the context of a trimeric complex (dotted arrow) in vivo. (**B**) Illustrative confocal capture of stage 10 embryo expressing Empty spiracles (Ems) fused to CC, together with CN-AbdA and VN-Exd fusion proteins, as indicated. BiFC is only occuring between AbdA and Ems, as expected from previous observation. (**C**) Illustrative confocal captures of stage 10 embryos expressing CN-AbdA, VN-Exd and ORF-CC constructs, as indicated in the different panels (Tin: Tinman; CtBP; Distalless, Dll; Bagpipe, Bag). Enlargements are provided in each case. White-dotted boxes depict nuclei where the ORF-CC interacts with both AbdA and Exd. Blue-dotted boxes depict nuclei where the ORF-CC interacts only with AbdA. Yellow-dotted boxes depict nuclei with absence of interaction. Fusion proteins are under the control of the *abdA-Gal4* driver. All expressing cells are recognized with the mCherry reporter. See also *Figure 6—figure supplements 1–3* and *Supplementary file 7*.
DOI: https://doi.org/10.7554/eLife.38853.020

The following figure supplements are available for figure 6:

**Figure supplement 1.** Illustrative confocal pictures of BiFC between ORF-VN and VC-Exd, as indicated.
DOI: https://doi.org/10.7554/eLife.38853.021

**Figure supplement 2.** Interaction properties of Extradenticle (Exd) with a set of 37 TFs that are positive with AbdA.
DOI: https://doi.org/10.7554/eLife.38853.022

**Figure supplement 3.** Adding the sfGFP to the fluorescence repertoire of the multicolour BiFC library.
DOI: https://doi.org/10.7554/eLife.38853.023

mCherry reporter) were positive for both green and blue fluorescent signals, thus demonstrating that simultaneous bicolor BiFC was efficient with different types of TFs in the live *Drosophila* embryo (*Figure 6C*). Interestingly, a close-up in the embryo revealed that several nuclei were only positive in the Cerulean channel, indicating a specific positive interaction status with AbdA (blue-dotted boxes in *Figure 6C*). Green-only nuclei could not be found in the case of the four positive TFs, although Exd is expressed at the same level as AbdA. This suggests that BiFC with Exd is dependent on the concomitant interaction with AbdA and could therefore not be revealed outside AbdA/Exd complexes. Few red-positive nuclei were also negative for the two fluorescent signals (as illustrated with the yellow box in the case of Bag-CC in *Figure 6C*), highlighting that the interaction with both AbdA and Exd was actively inhibited in these specific nuclei.

Together, these observations confirm that the ORF-CC library is compatible for performing bicolour BiFC, providing a unique opportunity to reveal and investigate cell-specific regulatory mechanisms of two different PPIs in vivo.

Since the CC fragment is compatible for doing BiFC with the N-terminal fragment of Venus and Cerulean, we tested whether it could also complement with the N-terminal fragment of an additional GFP-derived protein called super-folder GFP (sfGFP). BiFC with sfGFP is described to rely on the complementation property between a long N-terminal fragment of 214 residues (sfGFPN) and a short C-terminal fragment of 19 residues (sfGFPC, [*Zhou et al., 2011*]). In addition to having distinct spectral properties from Venus and Cerulean (*Figure 6*-figure 3), sfGFP has also a shorter maturation time (*Zhou et al., 2011*).

Here, we tested the complementation between sfGFPN and CC fragments by considering five TFs that were positive with Exd in our previous BiFC analyses. The sfGFPN fragment was fused at the N-terminus of Exd, making a sfGFPN-Exd fusion protein with the same fusion topology as previously done with the VN fragment (Materials and methods). BiFC was analyzed with excitation and emission wavelengths of the sfGFP. Results showed that the sfGFPN-Exd fusion protein could complement with the CC fragment in the case of all five tested TFs (*Figure 6—figure supplement 3*). Moreover, fluorescent signals were of similar brightness (*Figure 6—figure supplement 3*) and maturation time when compared to signals obtained with VN-Exd. Finally, since the sfGFPN and sfGFPC fragments are described as having a strong auto-affinity (*Zhou et al., 2011*), we also performed BiFC with the negative control Ems-CC. No fluorescent signal was obtained between Ems-CC and sfGFPN-Exd (*Figure 6—figure supplement 3*), confirming that the affinity between the sfGFPN and CC fragments is not strong enough to artificially induce complex formation between the two non-interacting proteins. Together these results establish yet another novel combination of fragment

complementation for BiFC, adding to the fluorescence repertoire that can be used with the multi-color BiFC library.

## Using the multicolor BiFC library to visualize PPIs in the overlapping endogenous expression domains

The multicolor BiFC library is under the control of UAS sequences. Pilot tests were performed as previously described (*Hudry et al., 2011*; *Baëza et al., 2015*), using a unique Gal4 driver that reproduces the expression domain of the Hox protein Ubx in the embryo (*Figure 7A*). The same rationale could be applied with a Gal4 driver reproducing the expression profile of the ORF (*Figure 7B*).

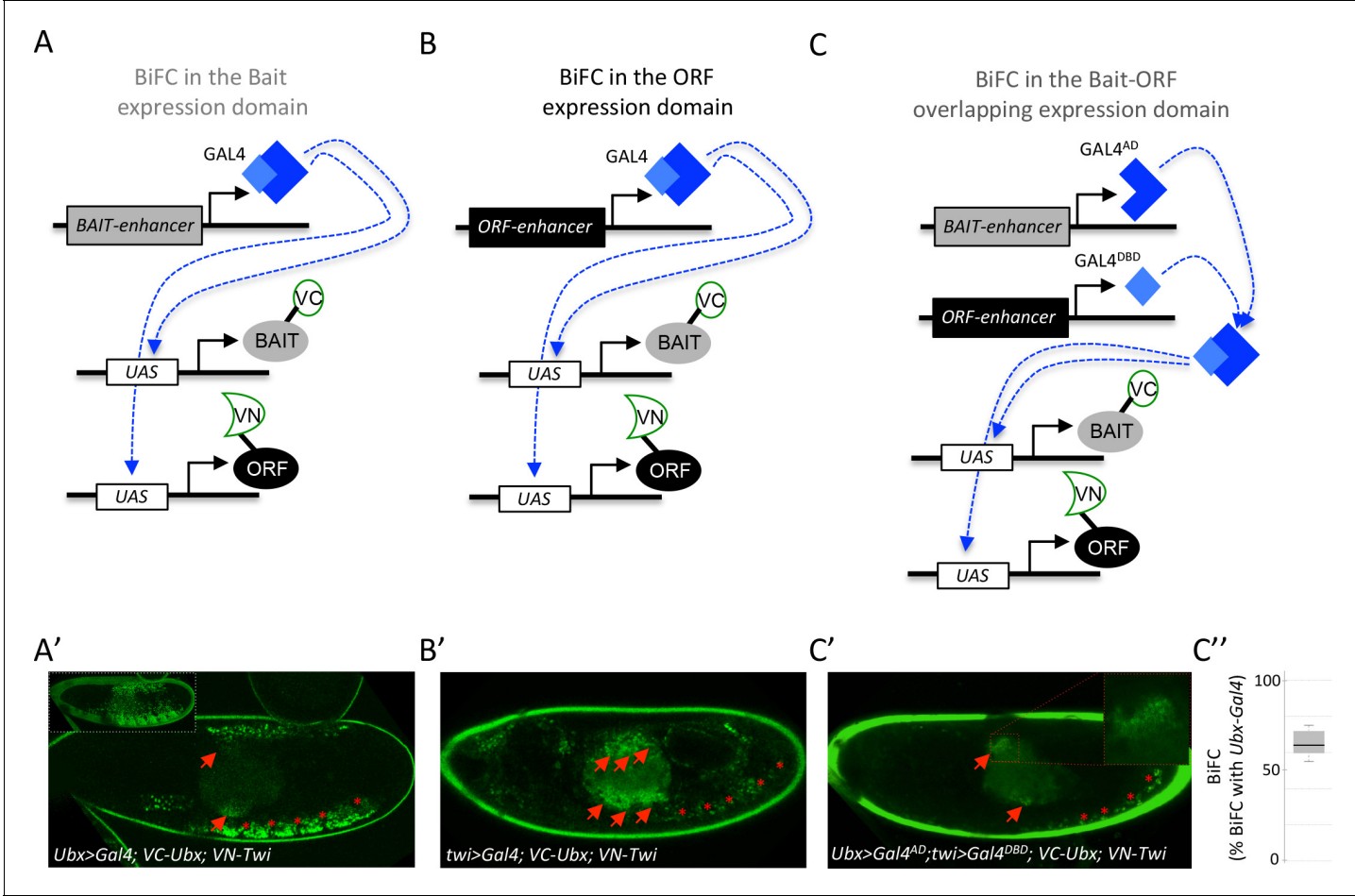

**Figure 7.** Coupling the multicolor BiFC library to the split-Gal4 system to visualize interactions in the overlapping expression domain of the two protein partners. (**A**) Principle of BiFC with a unique Gal4 driver reproducing the expression profile of the bait protein (for example *Hox-Gal4* driver). (**B**) Principle of BiFC with a Gal4 driver reproducing the expression profile of the ORF. (**C**) Principle of BiFC upon the independent expression of Gal4 moieties (Gal4AD and GALDBD) by using two different enhancers from the bait- or ORF-encoding gene. This system allows producing a functional Gal4 protein in the overlapping expression domain of the two enhancers, therefore assessing BiFC in cells that normally express both the bait and the ORF. (**A'**) Illustrative confocal picture of BiFC obtained upon the expression of Ultrabithorax (Ubx) and Twist (Twi) fusion proteins by using the *Ubx-Gal4* driver. Confocal acquisitions were specifically obtained at the level of the visceral mesoderm to better highlight BiFC signals in the midgut (red arrow). Red stars depict signals in the somatic mesoderm. Insert shows acquisition of BiFC signals in the epidermis. (**B'**) Illustrative confocal picture of BiFC obtained upon the expression of Ubx and Twi fusion proteins by using the *twi-Gal4* driver. Confocal acquisitions were specifically obtained at the level of the visceral mesoderm. BiFC is occurring in the entire visceral mesoderm of the midgut (red arrows). Fluorescence is also occurring in cells of the somatic mesoderm (red stars). (**C'**) Illustrative confocal picture of BiFC obtained upon the expression of Ubx and Twi fusion proteins by using the split-Gal4 (*Ubx-Gal4AD/twi-Gal4DBD*) system. BiFC is occurring in the same specific part of the midgut as in A' (enlargement on fluorescent nuclei is shown) and in few cells of the somatic mesoderm. C''. Quantification of the fluorescence intensity obtained in the visceral mesoderm with the slipt-Gal4 system (using the fluorescence intensity obtained in the same tissue with the *Ubx-Gal4* driver as a reference value). See also *Supplementary file 9*.
DOI: https://doi.org/10.7554/eLife.38853.024

Under these conditions, BiFC signals are visualized in the expression domain of only one interacting partner. Assessing BiFC in a context where the two candidate partners are expressed in their endogenous domains could therefore improve the confidence in the interpretation.

We have explored this direction by considering the split-Gal4 system (*Luan et al., 2006*), which consists in reconstituting an active Gal4 protein upon association of two separate Gal4-DNA-binding (Gal4$^{DBD}$) and Gal4-activation (Gal4$^{AD}$) domains. This property can be used to activate any UAS-driven gene in cells that will independently co-express the Gal4$^{DBD}$ and Gal4$^{AD}$ moieties, for example under the control of two different enhancers (*Luan et al., 2006*). We reasoned that the split-Gal4 system could be used to assess BiFC specifically in the overlapping expression domain of the two candidate partners (*Figure 7C*). This strategy implies having enhancers that could reproduce the expression profile of each candidate partner and that are available for Gal4$^{DBD}$ and/or Gal4$^{AD}$ expression. A number of enhancers have been used for this purpose in the Janelia fly line collection (https://www.janelia.org/project-team/flylight). In addition, Mimic transposons have also been designed to be compatible with the split-Gal4 system (*Venken et al., 2011*; *Nagarkar-Jaiswal et al., 2015*). Altogether these genetic tools allow BiFC to reproduce the expression of thousands of genes in *Drosophila* using the split-Gal4 system.

Here the suitability of the split-Gal4 system with BiFC was tested by considering the interaction revealed between Ubx and the mesodermal TF Twist (Twi) (*Baëza et al., 2015*). We used two enhancers from the Janelia collection that could recapitulate the expression profile of either *Ubx* or *twi* in the embryo. Each enhancer was first tested with the classical UAS/Gal4 system. Driving VC-Ubx and VN-Twi constructs with *Ubx-Gal4* led to prominent BiFC in the epidermis (insert in the *Figure 7A'*), but also to signals in the somatic mesoderm (red stars in the *Figure 7A'*) and in a specific part of the midgut (red arrows in the *Figure 7A'*). Expression of VC-Ubx and VN-Twi with the *twi-Gal4* driver led to BiFC in the entire midgut (red arrows in the *Figure 7B'*) and in cells of the somatic mesoderm (red stars in the *Figure 7B'*). The same *Ubx* and *twi* enhancers were engineered with the split-Gal4 system, and were used to respectively express the Gal4$^{AD}$ or Gal4$^{DBD}$ moiety. In this genetic context, BiFC signals could be detected in few cells of the somatic mesoderm (red stars in the *Figure 7C'*) and in a localized region of the midgut, as previously observed with the *Ubx-Gal4* driver (red arrows and highlighted in the enlargement of *Figure 7C'*). Signals obtained with the combined split-Gal4 system are weaker than signals obtained with each individual Gal4 driver, as exemplified in the midgut when comparing with *Ubx-Gal4* (*Figure 7C''*). This could be explained by the diminished activation activity of the reconstituted Gal4 protein when compared to intact Gal4 (*Luan et al., 2006*). In any case, this result confirms that the split-Gal4 system can be used with the multicolor BiFC library to analyze PPIs in the overlapping expression domain of the two interacting partners in vivo, allowing reproducing more closely the endogenous interaction.

## Discussion

### A ready-to-use fly library for analyzing the interactions of hundreds of TFs in vivo

We present a new fly line library called multicolor BiFC library that currently allows using 450 TFs for testing PPIs in vivo. This library contains two different sets of fly lines. The first set contains 232 TFs fused to VN and derives from previous work (*Hudry et al., 2011*; *Baëza et al., 2015*) or from swapping experiments with the original FlyORF library. Although swapping experiments were performed for TF-encoding ORFs, it should be noticed that the FlyORF library actually covers around 3000 ORFs and is therefore not limited to TFs only. Proof of principles described with the VN-swapping fly lines could apply to many more ORFs since BiFC is compatible with different types of proteins. Moreover, using the complementary VC-swapping fly line (*Bischof et al., 2013*) enables the full repertoire of the FlyORF library to be used for Venus-based BiFC in *Drosophila*.

The second set of the multicolor BiFC library consists of 326 ORF-CC constructs that are compatible with the first set of ORF-VN fly lines for doing Venus-based BiFC. ORF-CC fly lines are also compatible for doing Cerulean- or sfGFP-based BiFC with a CN- or sfGFPN-fused bait protein, respectively. This property allows multicolor BiFC experiments, which was here demonstrated with the analysis of interaction properties of different TFs in the context of the AbdA/Exd partnership.

The multicolour BiFC system provides an unprecedented opportunity to dissect cell-specific regulatory mechanisms involving three interacting proteins in vivo.

## BiFC library enables high-confidence large-scale interaction screens

Among the 260 TFs that have been used for the large-scale interaction screen, 35 were tested in fusion with VN or CC to assess for reproducibility of BiFC when using a different complementation strategy (*Supplementary file 8*). This analysis showed that only 2/35 TFs (Hr83 and Ravus) did not reproduce the same result with VC-AbdA or VN-AbdA (*Supplementary file 8*). This observation highlights that the fusion topology is of minor incidence (6%) for causing false negatives, suggesting that PPIs are rarely completely abolished by inappropriate BiFC fusion topologies. In addition, it should be noted that the AbdA (and Ubx) fusion proteins used in this study were originally generated with an exaggerated short linker region (three residues long) to better measure the potential influence of fusion topologies when establishing BiFC in *Drosophila* (*Hudry et al., 2011*). Using bait proteins fused to the Venus or Cerulean fragment with a more classical linker region (GGGSGG) could further diminish the influence of the fusion topology without increasing the risk of revealing indirect PPIs.

Our screen revealed that half of the 260 TFs could interact with Ubx or AbdA. As expected, a large proportion of the positive interactions (71%) were common to the two Hox proteins, which reflects their ability to control a number of identical developmental processes during embryogenesis. This hypothesis is supported by the observation that common interactions were not so frequent between Ubx and a more divergent Hox protein like Sex combs reduced (40% of common interactions among 35 tested TFs [*Baëza et al., 2015*]), or between AbdA and Exd (59% of common interactions among 37 tested TFs).

The diversity of positive TFs reflects the likely high potential of Hox proteins to engage in many different types of interactions in vivo, as previously shown for Ubx (*Bondos et al., 2006*). In addition, activators or repressors were equally enriched in Ubx and AbdA interactomes, highlighting that Hox proteins are involved in both positive and negative regulation of transcription. The screen was performed with a unique *Hox-Gal4* driver and therefore in conditions that do not reproduce the expression profile of the tested ORF. Despite this, we found that TFs that are endogenously more frequently co-expressed with Ubx and AbdA during embryogenesis were significantly enriched among the positive interactions.

Specificity of the BiFC screen was further demonstrated by the identification of several interactions that were exclusive to either Ubx or AbdA. Interestingly, the AbdA-specific interactome was significantly enriched in TFs that are expressed in tissues (gonad, heart) or cells (oenocytes) with known AbdA-specific functions.

Finally, the functional analysis of a selected pool of TFs expressed in the haltere disc showed that 80% of the interpretable phenotypes (25/31 TFs) were consistent with the interaction status found by BiFC in the embryo.

Altogether these results confirm that the multicolor BiFC library is suitable for performing large-scale interaction screens and revealing specific interactomes even for two closely related bait proteins.

## Suggestions for fine-tuning the specificity of the multicolor BiFC library screens

The multicolor BiFC library relies on the inducible UAS/Gal4 expression system, and caution should be taken regarding expression levels since high doses of protein expression could lead to artificial positive signals (*Kerppola, 2008*). Nevertheless, we recommend using a unique Gal4 driver in the case of a large-scale interaction screen. Ideally, this driver should derive from a *P-Gal4* insertion that will affect the expression of the bait-encoding gene, as done here with the *Ubx-Gal4* or *abdA-Gal4* driver. This genetic context allows expressing the bait protein at normal levels while eliminating one dose of endogenous competitive gene product. Furthermore, expressing the cofactor in cells that do not contain the corresponding endogenous gene product allows doing BiFC in the absence of competition, therefore increasing assay sensitivity. Here interactions were analyzed in the epidermis, which is an easily accessible tissue with low fluorescent background. The epidermis was previously shown to be tolerant for interactions with many different types of TFs (*Baëza et al., 2015*). The *Hox-*

*Gal4* expression domain is also large in the epidermis, making it an ideal tissue for a large-scale interaction screen. By comparison, the split-Gal4 system will not provide a better resolution since it will reproduce the expression pattern of only one cofactor with the bait protein. Therefore, not all potential interactors will be observed in their normal domain. In addition, the split-Gal4 system is less active and not compatible with *P-Gal4* insertions. In conclusion, we recommend using a bait-Gal4 driver derived from a *P-Gal4* insertion for performing a large-scale BiFC interaction screen and doing competition experiments (this work and [*Baëza et al., 2015*]) for validating the specificity of BiFC. Mutations affecting the interaction potential of the bait protein can also be used as a specificity control, as previously shown with mutated Hox proteins (*Baëza et al., 2015*), and the interaction confirmed by alternative methods such as co-immunoprecipitation (*Baëza et al., 2015*).

We showed that the interaction could be confirmed in the relevant developmental context in vivo by doing BiFC with the split-Gal4 system, In particular, the split-Gal4 system was used to visualize BiFC specifically in the overlapping expression domain of Ubx and Twi. The large number of fly lines compatible with the split-gal4 system makes this system very attractive for the future use of the multicolor BiFC library. Along the same line, the Janelia and Mimic collections propose hundreds of fly lines compatible with the LexAop/LexA-AD induction system. This expression system is also compatible with the multicolor BiFC library since UAS sequences of ORF-VN and ORF-CC constructs are swappable with LexAop sequences upon FLP/FRT-mediated recombination (*Bischof et al., 2014*). The two candidate partners could then be expressed independently and in their respective domain by using the UAS/Gal4 system on the one side, and the LexAop/Lex-AD system on the other side. In this context, BiFC will only be analyzed in the overlapping expression domain of the two partners, as described with the split-Gal4 system. Since the ORF-CC library is inserted on the second chromosome, we have generated another LexAop-swapping fly line that is compatible for FRT-mediated recombination with this chromosome (see Materials and methods).

In conclusion, the versatile inducible expression systems for thousands of enhancers in *Drosophila* and the multicolor BiFC library enable selection of biologically relevant conditions that better recapitulate the endogenous expression profile of each candidate partner, thus maximizing specificity and providing an unprecedented combination of possibilities for large-scale and/or in depth analysis of PPIs in a live animal organism.

## Materials and methods

### Plasmid constructions

#### Cloning of pTSVNm9short.attB
The FRT2-VNm9short fragment was PCR-amplified from plasmid pTSVNm9.attB with forward primer FRT2-VN-F (5'-TATGGATCCGAAGTTCCTATTCTCTACTTAGTATAGGAACTTCGATGGTGAG-CAAGGGCG-3') and reverse primer tub-R2 (5'-ACACTGATTTCGACGGTTACC-3'), thereby eliminating a stretch of 12aa between FRT2 and VNm9 and introducing flanking restriction sites BamHI and NotI. Plasmid pTSeGFP.attB (without the *yellow* marker insert) was digested with BamHI-NotI and the shortened FRT2-VNm9short fragment was inserted. Finally, the yellow marker gene was inserted via the XhoI site downstream of the fragment, resulting in pTSVNM9short.attB (for additional cloning details see [*Bischof et al., 2013*]).

#### Cloning of pTSCN.attB
The N-terminal part of Cerulean (CN) including the FRT2 site was PCR-amplified from the Cerulean cDNA (*Shyu et al., 2006*; *Hudry et al., 2011*) with forward primer FRT2-VN-F (5'-TATGGA TCCGAAGTTCCTATTCTCTACTTAGTATAGGAACTTCGATGGTGAGCAAGGGCG-3') and reverse primer Not-CN-R (5'-ATAGCGGCCGCctaGGTGATATAGACGTTGTCG-3'), thereby introducing BamHI and NotI sites. Note, the start sequence of the N-terminal end of Venus and Cerulean is identical. Subsequently, the CN fragment and the yellow marker were introduced into pTSeGFP.attB the same way as described above, generating plasmid pTSCN.attB.

#### Cloning pGW-CC.attB
To generate this plasmid, the 3 HA tag from pGW-HA.attB (*Bischof et al., 2013*) was replaced with the C-terminal portion of Cerulean (CC). In brief, fragment CC was PCR-amplified from the Cerulean

cDNA with primers CC-F (5'-ATAGGTACCTGCCGACAAGCAGAAGAACG-3') and CC-R (5'-TATGC TAGCTTACTTGTACAGCTCGTCCATGCCG-3'). Note, in this construct no FRT2 sequence was included in front of the Cerulean fragment, as no swapping experiments are intended with the subsequent fly lines resulting from this construct. The CC fragment was inserted into pGW-HA.attB with KpnI-NheI digestion, thus releasing FRT2 fragment and 3xHA tag and generating plasmid pGW-CC. attB.

### Creating the TF-ORF-CC plasmid library
A detailed description of the involved Gateway cloning steps to generate such a library can be found in previous publications (*Bischof et al., 2013*; *Bischof et al., 2014*). In brief, the *Drosophila* transcription factor collection in the pDONR221 vector (*Hens et al., 2011*) was used as a source for the ORFs. From this collection the TFs were shuttled (Gateway subcloned) into the *Drosophila* expression vector pGW-CC.attB, thus fusing the ORF to the CC tag and equipping them with UAS regulatory promoter elements.

### Cloning of sfGFPN-Exd in pUASTattB
The N-terminal fragment of sfGFP (sfGFPN : residues 1–214 [*Zhou et al., 2011*]) was PCR amplified from the sfGFP cDNA and cloned in place of VN in the original VN-Exd construct in pUASTattB between EcoRI and XhoI restriction sites (*Hudry et al., 2011*).

## Generation of fly strains
For all germline transformation experiments we used the ΦC31 integrase-mediated site-specific integration method.

### Generation of individual fly strains for promoter or tag swapping
*TSVNm9short-86Fb, TSCN-86Fb* strains: The constructs pTSVNM9short.attB and pTSCN.attB were injected into line *ΦX-86Fb*. Transgenic offspring was made homozygous for these transgenes and combined with an X chromosome-linked hsp70-flp construct. These strains provide either the Venus N-terminal tag or the Cerulean N-terminal tag for a swapping event at the 3'end of any UAS-ORF strain from FlyORF that is based on cloning with pGW-HA.attB.

For the *PSlexO-21F* strain the pPSlexO.attB plasmid was injected into line *ΦX-21F*, again followed by balancing the strain for the transgene and providing the hsp70-flp construct on the X-chromosome. This strain can be used to exchange the UAS-regulated promotor of the TF-ORF-CC library (at location 21F) for a lexA operator (lexO) promotor (*Bischof et al., 2013*; *Bischof et al., 2007*).

### Creating a transcription factor cerulean tagged fly library (UAS-ORF-CC) on the second chromosome
The individual UAS-ORF-CC plasmids were combined in small pools and injected into *ΦX-21F*. Transgenes that were not recovered after outcrossing and PCR identification were re-pooled and injected again. The PCR identification for this library was done by single-fly PCR and Sanger sequencing into the 5' region of the genes. Individual stocks were created with balancer line (y⁻w⁻; Bc gla/SM6 a). Pool injection, transgene identification and stock generation were described in detail previously (*Bischof et al., 2013*; *Bischof et al., 2014*).

### Swapping procedures: exchange of 3xha tag for VNm9 or VNm9short
The presence of the mutated FRT2 site downstream of the ORF was used to replace the 3xHA tags of FlyORF lines for the Venus tags VNm9 or VNm9short. In brief, the ORF-3xHA lines were crossed to flies carrying a swap construct together with the hsp-flp transgene. After heat-shock treatment and outcrosses the swapping events (FLP/FRT-mediated in vivo events) can easily be tracked by screening for a specific marker combination, i.e. y + w+ for a C-terminal tag exchange. This procedure was described in detail previously (*Bischof et al., 2013*; *Bischof et al., 2014*).

### Fly lines availability

Fly lines generated for the project will be deposited in the FlyORF library and are available upon request to FlyORF (https://flyorf.ch/index.php/orf-collection)

## BiFC analysis

Fly crosses, embryo preparation and BiFC observation in live embryos were performed as previously described (*Hudry et al., 2011*; *Baëza et al., 2015*). Quantification of BiFC in the *Figure 7C''* was specifically performed in the visceral mesoderm. Briefly, observations were done at least two or three times (from two different overnight egg laying periods) with wild-type or mutated Hox proteins, respectively. BiFC signals with wild type Hox proteins were considered as positive when the intensity was above the fluorescent background and reproducible in the expected proportion of embryos from independent preparations (around 200 embryos were mounted in each case for confocal observation). Fluorescence intensities are strongly fluctuating from one TF to another, which could be due to the influence of the fusion topology and/or a real variation in interaction affinity with the Hox protein. Identical parameters of confocal acquisition were applied with mutated Hox proteins or in competition experiments.

## Fly stocks

The *Ubx-Gal4* and *abdA-Gal4* lines were previously used (*Hudry et al., 2011*). UAS-ORF-HA fly lines used for swapping and competition experiments are from the FlyORF library (*Bischof et al., 2013*). *Ubx-Gal4$^{AD}$* and *twi-Gal4$^{DBD}$* are from Janelia (Bloomington stock numbers 70646 and 68953 respectively). RNAi against TFs are from Bloomington and were expressed with the *MS1096* driver combined or not with UAS-dicer in individuals heterozygous for the *abxpbxbx* mutation, which affects *Ubx* expression in the haltere disc (*de Navas et al., 2011*). The transgenic BiFC library lines will be available from FlyORF (http://www.flyorf.ch).

## Statistical analyses

Wilcoxon statistical tests were performed by using Python and R in-house scripts. Quantification of ectopic wing-like hairs in the haltere was shown by boxplot representation using RSoftware. Boxplot depicts the smallest value, lower quartile, median, upper quartile, and largest value for each condition.

## Acknowledgements

We thank the Bloomington stock center for fly lines and Bart Deplancke for sharing the TF ORF plasmid library and Cristina Bastos for assistance with injections. Work in the laboratory of S Merabet was supported by Association pour la Recherche sur le Cancer (ARC, PJA20141202007), Fondation pour la Recherche Médicale (FRM, DEQ20170336732), Ligue Nationale Contre le Cancer, Centre National de Recherche Scientifique (CNRS), CEFIPRA (5503–2), and Ecole Normale Supérieure (ENS) de Lyon. Work in the laboratory of K Basler was supported by the Swiss National Science Foundation.

## Additional information

### Competing interests

Johannes Bischof, Konrad Basler: Involved in maintaining and distributing the fly lines via the not-for-profit FlyORF project. There are no other competing interests to declare. Mikael Björklund: Involved in the development of the FlyORF resource. There are no other competing interests to declare. The other authors declare that no competing interests exist.

## Funding

| Funder | Grant reference number | Author |
|---|---|---|
| Fondation pour la Recherche Médicale | 1122556 | Johannes Bischof<br>Marilyne Duffraisse<br>Edy Furger<br>Leiore Ajuria<br>Guillaume Giraud<br>Solene Vanderperre<br>Rachel Paul<br>Samir Merabet |
| Indo-French Centre for the Promotion of Advanced Research | | Johannes Bischof<br>Marilyne Duffraisse<br>Edy Furger<br>Leiore Ajuria<br>Guillaume Giraud<br>Solene Vanderperre<br>Rachel Paul<br>Samir Merabet |

The funders had no role in study design, data collection and interpretation, or the decision to submit the work for publication.

## Author contributions

Johannes Bischof, Conceptualization, Resources, Data curation, Formal analysis, Investigation, Methodology, Writing—review and editing; Marilyne Duffraisse, Data curation, Validation, Investigation, Visualization; Edy Furger, Resources, Data curation, Formal analysis, Methodology; Leiore Ajuria, Resources, Formal analysis, Validation, Investigation, Visualization, Methodology; Guillaume Giraud, Solene Vanderperre, Rachel Paul, Formal analysis, Visualization; Mikael Björklund, Resources, Data curation, Investigation, Writing—review and editing; Damien Ahr, Investigation, Visualization; Alexis W Ahmed, Lionel Spinelli, Christine Brun, Formal analysis, Methodology; Konrad Basler, Supervision, Funding acquisition; Samir Merabet, Conceptualization, Data curation, Formal analysis, Supervision, Funding acquisition, Validation, Investigation, Visualization, Methodology, Writing—original draft, Project administration, Writing—review and editing

## Author ORCIDs

Mikael Björklund (iD) http://orcid.org/0000-0002-2176-681X
Christine Brun (iD) https://orcid.org/0000-0002-5563-6765
Samir Merabet (iD) http://orcid.org/0000-0001-7629-703X

## Decision letter and Author response

Decision letter https://doi.org/10.7554/eLife.38853.036
Author response https://doi.org/10.7554/eLife.38853.037

## Additional files

### Supplementary files

• Supplementary file 1. List of the 235 TFs of the multicolor BiFC library fused to the VN fragment at the C- or N-terminus, with a long or short linker region, as indicated. Note that several TFs are fused to VN with either a long or a short linker region.
DOI: https://doi.org/10.7554/eLife.38853.025

• Supplementary file 2. List of the 326 TFs of the multicolor BiFC library fused to the CC fragment. TFs highlighted in light green are also available as VN fusion constructs (see *Supplementary file 1*).
DOI: https://doi.org/10.7554/eLife.38853.026

• Supplementary file 3. List of the 260 VN- and CC-fusion TFs of the multicolor BiFC library that were tested with Ubx and AbdA.
DOI: https://doi.org/10.7554/eLife.38853.027

• Supplementary file 4. Expression profile of the 260 TFs that were tested with Ubx and AbdA among 25 different developmental contexts of the *Drosophila* embryo. Each color code corresponds

to a different tissue when expressed. Black boxes depict expression in tissues where Ubx and AbdA are not present. The last two columns indicate the number of co-occurrences of the TF and the Hox protein with regard to the total distribution (parentheses).
DOI: https://doi.org/10.7554/eLife.38853.028

• Supplementary file 5. Analysis of the proportion of common and specific AbdA- and Ubx interactors expressed in the different tissues, as indicated.
DOI: https://doi.org/10.7554/eLife.38853.029

• Supplementary file 6. List of TFs tested in RNAi in Ubx heterozygous mutant haltere discs. Green and red boxes correspond to increased or not increased RNAi phenotypes, respectively. Black boxes correspond to RNAi phenotypes that could not be interpreted with regard to a potential Ubx cofactor function in the haltere disc (due to morphogenesis defects). Stars indicate RNAi fly lines that were tested with dicer. Stocks without the star are from the last generation and were tested without dicer.
DOI: https://doi.org/10.7554/eLife.38853.030

• Supplementary file 7. List of VN fusion TFs that were positive in BiFC tests with VC-AbdA and tested with VC-Exd. Green and red boxes indicate a positive or negative interaction status, respectively.
DOI: https://doi.org/10.7554/eLife.38853.031

• Supplementary file 8. List of the 35 TFs tested as VN or CC fusion constructs with VC-AbdA or VN-AbdA, respectively. Yellow boxes highlight the two TFs that showed opposite interaction status in the context of the two different fusion topologies (Hr83 and Ravus).
DOI: https://doi.org/10.7554/eLife.38853.032

• Supplementary file 9. Quantification of BiFC resulting from the overlap between twi- and Ubx-Gal4 drivers in the visceral mesoderm.
DOI: https://doi.org/10.7554/eLife.38853.033

• Transparent reporting form
DOI: https://doi.org/10.7554/eLife.38853.034

## Data availability

Fly lines generated for the project have been deposited to the FlyORF library and are available upon request to FlyORF (https://flyorf.ch/index.php/orf-collection). The numerical, processed data used for this study is provided in the manuscript, figures and supplementary files.

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
