## [Decision Letter]

Thank you for submitting your article "Generation of a versatile BiFC ORFeome library for analyzing protein-protein interactions in live *Drosophila*" for consideration by *eLife*. Your article has been reviewed by two peer reviewers, and the evaluation has been overseen by K VijayRaghavan as the Senior/Reviewing Editor. The reviewers have opted to remain anonymous.

The reviewers have discussed the reviews with one another and the Reviewing Editor has drafted this decision to help you prepare a revised submission.

Summary:

In this manuscript, a new resource is presented, a set of *Drosophila* fly lines called "multicolor BiFC library", which covers most of the *Drosophila* transcription factors and thus allows to capture transcription factor (TF) interactions using the Bimolecular Fluorescence Complementation (BiFC) in live conditions. Importantly, the multicolor BiFC can be used to simultaneously probe two binary interactions, is compatible for large-scale interaction screens and can be combined with the existing *Drosophila* tools, for example, the split-GAL4 system and thus allows a visualization of potential interactions only in domains where the proteins of interest are co-expressed.

Overall, the manuscript is very solid and provides an important genetic toolbox for the identification and analysis of protein-protein interactions. Although BiFC has a couple of disadvantages, this resource is nonetheless very valuable. Our initial enthusiasm on reading the manuscript was somewhat tempered on careful reading. Several important issues need to be addressed before publication in *eLife* and these are listed below.

Essential revisions:

1) The authors have described before the usability of BiFC to monitor interactions in vivo. What is new and exciting in this paper is the use of this system to show interactions of not only two but also of three interaction partners. The authors show that they can detect interactions between Abd-A and a third partner and Exd and the same third partner (Figure 6). However, the co-occurrence of blue and green signals in the same cell is not proof that the three proteins appear in a complex. These could be simply independent interactions in the same cell. The issue of whether using BiFC to study interactions between three proteins provides information about tripartite complexes is important. The authors appear to be aware of the issue that the assay only directly reports on binary interactions, describing "analyzing two different PPIs simultaneously" and in Figure 6 legend "potentially in the context of…complexes". The relationship between simultaneous reporting of two binary interactions and information about tripartite complexes should be clarified. The authors should drop the term "tripartite" from the Abstract and instead generally make it clear that the advance is to study two binary interactions in the same cell. This may provide information on potential complexes and they should present how the binary information might be used; e.g. where they suggest that "BiFC with Exd is dependent on the concomitant interaction with AbdA" since TF-Exd BiFC (green) is only found in cells that also have TF-AbdA BiFC (blue). They should also remove in Figure 6A the 'tripartite complex' or indicate somehow that this is just a possibility but not shown in the paper.

2) For showing the functional relevance of the identified interactions, the authors used the haltere-to-wing transformation and a sensitized mutant background (Ubx heterozygous condition). This is an elegant experiment, however, the pictures in Figure 5 are not good enough. The authors need to show higher magnifications and better images of the bristles. In addition, quantification of the phenotypes for all genotypes is needed.

3) The authors introduce the super-folded GFP as an additional GFP derived protein useable for BiFC and show that this variant behaves similarly to VN fusions. Their reasoning for introducing this variant was that it has shorter maturation times and distinct spectral properties from Venus and Cerulean. However, the authors do not specifically test the usability of this variant for detecting interactions with unstable proteins. Thus, the data as presented just show that the sfGFP variant works similarly well as the VN variants for more stable proteins, which is good but not surprising. If the authors could show an interaction using an unstable TF comparing the performance of the Venus and sfGFP variant, this would strengthen this point very much. In any case, the authors should move the sfGFP part to the supplement if they cannot show its usability for a more unstable protein.

4) The use of the split-GAL4 system is an excellent addition, however, it also raises the question why the strong interaction of Ubx and Twi in the visceral mesoderm, which is clearly visible with the split-GAL4 system (8C') is not detectable with the *Ubx-GAL4* or *twi-GAL-4* drivers (8A', C'). Careful quantification of the fluorescence signal is required for the split-GAL4 line described in Figure 8, as the fluorescence intensities are quite different. Does this result also mean that BiFC should be in principle only be done with the split-GAL4 system? This needs to be carefully discussed.

5) Figure 3 is unreadable. Could the authors think of better ways to represent the interactomes? For example, they could show the overlap of Abd-A and Ubx interactors in a circle diagram etc. They show that HD TF are enriched, any other class? What about the function of these TFs: repressors, activators? The same is true for Figure 4, the information could be better and more intuitively presented.

6) The paper could be considerably shortened. The presentation of the large-scale BiFC analysis of interactions with Ubx and AbdA (subsection “Using the multicolor BiFC library for large-scale interaction screens in live *Drosophila* embryos”) is followed by a number of approaches seeking to "validate" the observed interactions. However, these approaches do not provide straightforward validation, demonstrating corroboration based on expected outcomes, but rather lead to novel findings. Although these novel findings, such as the difference in IDR and SLiM occurrence between Ubx- and AbdA-specific interactomes and differential effects of HX mutations on Ubx and AbdA, are intriguing, they do not provide clear validation of the BiFC observations. We would recommend deleting the sections beginning "The multicolor BiFC library reveals Hox-specific…" and "A short conserved Hox protein motif…".

---

## [Author Response]

Essential revisions:1) The authors have described before the usability of BiFC to monitor interactions in vivo. What is new and exciting in this paper is the use of this system to show interactions of not only two but also of three interaction partners. The authors show that they can detect interactions between Abd-A and a third partner and Exd and the same third partner (Figure 6). However, the co-occurrence of blue and green signals in the same cell is not proof that the three proteins appear in a complex. These could be simply independent interactions in the same cell. The issue of whether using BiFC to study interactions between three proteins provides information about tripartite complexes is important. The authors appear to be aware of the issue that the assay only directly reports on binary interactions, describing "analyzing two different PPIs simultaneously" and in Figure 6 legend "potentially in the context of…complexes". The relationship between simultaneous reporting of two binary interactions and information about tripartite complexes should be clarified. The authors should drop the term "tripartite" from the Abstract and instead generally make it clear that the advance is to study two binary interactions in the same cell. This may provide information on potential complexes and they should present how the binary information might be used; e.g. where they suggest that "BiFC with Exd is dependent on the concomitant interaction with AbdA" since TF-Exd BiFC (green) is only found in cells that also have TF-AbdA BiFC (blue). They should also remove in Figure 6A the 'tripartite complex' or indicate somehow that this is just a possibility but not shown in the paper.

We agree that “tripartite complex” is an over-interpretation and we have removed the term form the Abstract. We now present this aspect as a possibility, both in the new Figure 6A (with a dotted arrow) and in the corresponding legend.

2) For showing the functional relevance of the identified interactions, the authors used the haltere-to-wing transformation and a sensitized mutant background (Ubx heterozygous condition). This is an elegant experiment, however, the pictures in Figure 5 are not good enough. The authors need to show higher magnifications and better images of the bristles. In addition, quantification of the phenotypes for all genotypes is needed.

Illustrative phenotypes of the different categories were de novo acquired with a scanning electron microscope and higher magnifications of ectopic bristles are also shown in the new Figure 5. Statistical quantification of the number of ectopic bristles is also provided for each positive RNAi (together with one representative negative RNAi, wild type and *abxpbxbx* background).

3) The authors introduce the super-folded GFP as an additional GFP derived protein useable for BiFC and show that this variant behaves similarly to VN fusions. Their reasoning for introducing this variant was that it has shorter maturation times and distinct spectral properties from Venus and Cerulean. However, the authors do not specifically test the usability of this variant for detecting interactions with unstable proteins. Thus, the data as presented just show that the sfGFP variant works similarly well as the VN variants for more stable proteins, which is good but not surprising. If the authors could show an interaction using an unstable TF comparing the performance of the Venus and sfGFP variant, this would strengthen this point very much. In any case, the authors should move the sfGFP part to the supplement if they cannot show its usability for a more unstable protein.

There is some confusion about the usability of sfGFP-based BiFC for detecting interactions with unstable proteins. The sfGFP protein has a faster maturation time than other GFP-derived proteins, and is therefore more appropriate for monitoring protein dynamics. However, BiFC with sfGFP- or Venus-derived fragments requires several hours for maturation, which forbids tracking protein interaction dynamics. To date, there is no fluorescent protein with a complementation maturation time that could be compatible with the analysis of protein interaction dynamics in vivo. In addition, sfGFP- or Venus-based BiFC stabilizes the interaction once it is revealed. In conclusion, the novel combination of fragment complementation between the sfGFPN and CC fragments just adds to the fluorescence repertoire of the multicolor BiFC library. It will not provide access to protein stability or interaction dynamics. We tried to reformulate more clearly this aspect to avoid any confusion with this regard. The corresponding results are also included in the supplement (Figure 6—figure supplement 16), as suggested by reviewers.

4) The use of the split-GAL4 system is an excellent addition, however, it also raises the question why the strong interaction of Ubx and Twi in the visceral mesoderm, which is clearly visible with the split-GAL4 system (8C') is not detectable with the Ubx-GAL4 or twi-GAL-4 drivers (8A', C'). Careful quantification of the fluorescence signal is required for the split-GAL4 line described in Figure 8, as the fluorescence intensities are quite different. Does this result also mean that BiFC should be in principle only be done with the split-GAL4 system? This needs to be carefully discussed.

Because of this interesting comment on the split-Gal4 system, we decided to better describe the interaction profile with each different driver. The use of the split-Gal4 system with the multicolor BiFC library was also more carefully discussed. Practically, the *Ubx-* and *Twi-Gal4* drivers lead to expression and interaction of Ubx and Twi in different parts of the embryo, including the visceral mesoderm. BiFC in the visceral mesoderm was not obvious in the previous Figure 8 of the first version, due to strong fluorescent signals in the epidermis or somatic mesoderm, respectively. We now present maximum intensity projections of confocal sections centered on the visceral mesoderm to better highlight BiFC in this tissue with each individual driver (A’ and B’ in the new Figure 7). In addition, we quantified the intensity of BiFC in the visceral mesoderm with the split-Gal4 system, in comparison to the BiFC intensity observed in the same tissue with the *Ubx-Gal4* driver. Values are in accordance with the known diminished activation activity of the reconstituted Gal4 when compared to full length Gal4 (which is now mentioned in the main text). Finally, we discussed why and how the split-Gal4 system should be used with the multicolor BiFC in the Discussion section (briefly, not for large-scale interaction screens, but rather for confirming/analyzing individual binary interactions in the relevant developmental tissue).

5) Figure 3 is unreadable. Could the authors think of better ways to represent the interactomes? For example, they could show the overlap of Abd-A and Ubx interactors in a circle diagram etc. They show that HD TF are enriched, any other class? What about the function of these TFs: repressors, activators? The same is true for Figure 4, the information could be better and more intuitively presented.

Interactomes of Ubx and AbdA are now presented differently, in a way that each TF name could be read upon higher magnification in the new Figure 4—figure supplement 4 and Figure 4—figure supplement 10. As suggested by reviewers, interactomes are also presented in a simplified manner to better illustrate the repartition of the main TFs families in the different Ubx and AbdA interactomes (new Figure 4 and Figure 4—figure supplement 11). Of note, the distribution of the different BiFC fly lines was voluntary switched from the supplement to the new Figure 3 to directly provide the global information that will evolve in the future.

6) The paper could be considerably shortened. The presentation of the large-scale BiFC analysis of interactions with Ubx and AbdA (subsection “Using the multicolor BiFC library for large-scale interaction screens in live Drosophila embryos”) is followed by a number of approaches seeking to "validate" the observed interactions. However, these approaches do not provide straightforward validation, demonstrating corroboration based on expected outcomes, but rather lead to novel findings. Although these novel findings, such as the difference in IDR and SLiM occurrence between Ubx- and AbdA-specific interactomes and differential effects of HX mutations on Ubx and AbdA, are intriguing, they do not provide clear validation of the BiFC observations. We would recommend deleting the sections beginning "The multicolor BiFC library reveals Hox-specific…" and "A short conserved Hox protein motif…".

The entire parts dedicated to the analysis of intra-molecular features of Ubx and AbdA interactors have been deleted according to the reviewers’ suggestion.